# Inactivation of Soybean Trypsin Inhibitor by Dielectric-Barrier Discharge Plasma and Its Safety Evaluation and Application

**DOI:** 10.3390/foods11244017

**Published:** 2022-12-12

**Authors:** Ye Xu, Yu Sun, Kunlun Huang, Jingjing Li, Chongshan Zhong, Xiaoyun He

**Affiliations:** 1College of Food Science and Nutritional Engineering, China Agricultural University, Beijing 100083, China; 2Lanzhou Anning District Bureau of Statistics, Lanzhou 730070, China; 3College of Information & Electrical Engineering, China Agricultural University, Beijing 100083, China

**Keywords:** dielectric-barrier discharge plasma, soybean trypsin inhibitor, inactivation, safety evaluation, soymilk quality, caco-2 cell viability, physicochemical properties, rheological properties

## Abstract

The trypsin inhibitor (TI) is one of the most important anti-nutritive elements in soybeans. As a new nonthermal technology, dielectric-barrier discharge (DBD) cold plasma has attracted increasing attention in food processing. In this research, we investigated the effect of dielectric-barrier discharge (DBD) plasma treatment on soybean trypsin inhibitor content and its structure, evaluated TI toxicity and the safety of its degradation products after treatment with DBD technology in vitro and in vivo, and applied the technology to soybean milk, which was analyzed for quality. Using the statistical analysis of Student’s *t*-test, the results demonstrated that DBD plasma treatment significantly decreased the content of TI (33.8 kV at 1, 3, or 5 min, *p* < 0.05, *p* < 0.01, *p* < 0.001) and destroyed the secondary and tertiary structures of TI. TI was toxic to Caco-2 cells and could inhibit body weight gain, damage liver and kidney functions, and cause moderate or severe lesions in mouse organ tissues, whereas these phenomena were alleviated in mice treated with degradation products of TI after DBD plasma treatment under the optimal condition (33.8 kV at 5 min). The content of TI in DBD-treated soymilk was also significantly reduced (*p* < 0.001), while the acidity, alkalinity, conductivity, color, and amino acid composition of soymilk were not affected, and there were no statistical differences (*p* > 0.05). In summary, DBD plasma is a promising non-thermal processing technology used to eliminate TI from soybean products.

## 1. Introduction

Soybean is now one of the most economically significant legume crops worldwide and also serves as the principal resource of plant protein and vegetable oil for humans [1]. Consumption of soybeans is increasing on account of reported advantageous influences on human health, including lowering plasma cholesterol, preventing cancer, diabetes, and obesity, and protecting against intestinal diseases [2]. Despite its numerous favorable nutritional properties, soybean is composed of various anti-nutritional bioactive compounds (lectins, protease inhibitors, etc.) that show adverse physiological effects hindering their utilization or digestibility. Among these anti-nutritional factors, trypsin inhibitors (TIs) are the major components that lessen the digestion and absorption of dietary proteins by restraining the digestive enzymes [3]. More specifically, in soybeans, TIs exert their function by forming stable stoichiometric complexes with the digestive enzymes trypsin and chymotrypsin, and the produced noncovalent complexes render the proteases inactive and significantly reduce the digestibility and utilization of proteins and amino acids by non-ruminants [4]. Therefore, to ensure the safety and nutritional quality of soybean foods, it is necessary to take effective measures to inactivate anti-nutritional factors during soybean processing.

As the most commonly used method of deactivation at present, the thermal processing method can kill microorganisms in food and inactivate enzymes, though if the processing temperature is too high or the time is too long, this may result in chemical or physical changes in food, thus affecting the sensory characteristics of food, reducing the activity or utilization of bioactive substances in food, and even producing harmful substances such as acrylamide and furan that cause chronic human diseases. Therefore, the current food processing technology mainly focuses on non-thermophysical processing technology, which maximizes the passivation of harmful substances in food, minimizes the destruction of the original nutrients in food, and maintains the biological and sensory characteristics of food. Emerging non-thermal food processing technologies include high pressure, ultrasound, pulsed electric field, plasma, etc. [5].

Cold plasma is an emerging non-thermal processing technology with a relatively low temperature (about 30~60 °C), which has become a very promising alternative to traditional thermal treatment technology in the field of food processing in terms of pesticide decontamination, enzyme inactivation, and toxin removal [6,7,8,9]. Cold plasma can be characterized by its generation methods, including atmospheric pressure plasma jet (APPJ), dielectric-barrier discharge (DBD), corona discharge (CD), and gliding arc discharges [10], among which DBD plasma is the most prevalent technology in the food industries [11]. DBD can directly treat the sample powder without preprocessing and modify macromolecular substances without adding exogenous substances, possessing the strengths of availability, safety, and greenness [12]. Liu et al. found that the immunoglobulin G binding capacity of β-lactoglobulin was decreased by 58.21% after 4 min of DBD plasma treatment, which was confirmed by Western blot and ELISA analysis [13]. Guo et al. discovered the use of DBD plasma could be a simple and green method to enhance the physicochemical properties of potato starch and its film [14]. Recent research on the inactivation of anti-nutritional factors in soybean by DBD by Liu et al. used DBD plasma treatment (40 kV, 12 kHz at 1, 2, 3, and 4 min) to eliminate soybean agglutinin (SBA) activity in a model of SBA system and soymilk, and found that the SBA in the model system and hemagglutination activity was reduced by 87.31%, and its inactivation was confirmed in soymilk [15]. Additionally, only one report suggested that DBD plasma could induce the inactivation of TI in soymilk and Kunitz-type TI from soybean in a model system [16]. Nevertheless, studies on the effects of plasma on soybean TI are still finite. Additionally, further studies are not only requisite to demonstrate the effects of DBD plasma treatment on the nutritive value and sensory quality of foods, but the safety of DBD-plasma-treated foods should also be comprehensively evaluated before plasma is extensively applied in the food industry. 

Some studies involving the application of DBD plasma in soymilk can be found; yet few studies deal with soybeans and soy flour. Therefore, the purpose of this study was to investigate the mechanism of trypsin inhibitor inactivation in soybean after soaking treatment or DBD plasma treatment at different times and to determine its optimal passivation conditions. A combined in vivo and in vitro safety evaluation model was developed to evaluate the safety of TIs and their degradation products by DBD plasma treatment. Moreover, the feasibility of DBD plasma treatment in soymilk processing was discussed from the aspects of physicochemical properties, rheological properties, and color of soymilk, which provided a theoretical basis for the application of non-heat treatment methods in soymilk industrial processing.

## 2. Materials and Methods

### 2.1. Materials

Soybeans (variety: Dongnong 252, moisture content 8.3%, crude protein content 42.87%, crude fat content 20.06%) were purchased from Dong Xiaonong Black Potato Workshop (Harbin, China). Human colorectal cancer cells (Caco-2) were purchased from the Institute of Basic Medicine, Chinese Academy of Medical Sciences (Beijing, China); trypsin inhibitor was purchased from Yuanye Biotechnology Co., Ltd. (Shanghai, China). In the experiment, 4-week-old SPF CD-1 male mice were selected as the animal subjects, purchased from Beijing Vital River Laboratory Animal Technology Co., Ltd. (Beijing, China). The experimental design was approved by the Laboratory Animal Welfare and Animal Experiment Ethics Review Committee of China Agricultural University. The quality certificate number is SCXK (Beijing) 2016-0006, and the animal experiment facility certificate number is SYXK (Beijing) 2020-0052. The maintenance feed was purchased from Beijing Keao Xieli Feed Co., Ltd. (Beijing, China) (experimental animal feed license number: SCXK (Beijing) 2015-0013).

### 2.2. Pretreatment

We accurately weighed a certain amount of soybeans, washed them three times, and soaked them with deionized (DI) water in a ratio of 1:10 for 16 h. After draining the water, we peeled the soybeans and placed them in an oven at 45 °C for drying. After drying, we used a small grinder to pulverize the soybeans for 90 s. To avoid overheating, we paused every 30 s. After pulverization, we sieved to obtain soybean powder and stored at −20 °C for analysis. Meanwhile, a certain amount of soybean was weighed, soaked for 16 h, and peeled. We accurately weighed 40 g of soaked soybeans, added 400 mL of distilled water, placed in a beater for 3 min, and filtered with double-layer nylon cloth to obtain untreated soymilk samples.

### 2.3. DBD Plasma Treatment

Soybeans were soaked, peeled, dried, crushed, and sieved, and then placed in a polyethylene Petri dish in an insulating container. The plasma processing apparatus included a high-voltage (0–50 kV) AC power supplied with a frequency range of 5–10 kHz by Nanjing Suman Electronics Co., Ltd. (Nanjing, China), two stainless steel electrodes, an oscilloscope (TPS2024B, Tektronix Co., Beaverton, OR, USA) with a high-voltage probe (P6015A, Tektronix Co.). Plasma was generated with a 10 mm gap between two stainless steel electrodes. The samples were placed and processed between stainless steel electrodes, the upper electrode of which was covered by a three-layer insulating plastic sheet, and the air was used as the feeding gas. The processing voltage was 33.8 kV (peak-to-peak), and the samples were treated for 1 min, 3 min, and 5 min, respectively. After the treatment, the samples were stored at −20 °C for follow-up determination. Subsequently, 5 mL of soymilk was placed in an insulating polyethylene plastic Petri dish and treated with a DBD plasma device (33.8 kV) for 5 min to obtain DBD-treated soymilk.

### 2.4. Determination of TI Activity

The method of measuring TI activity of the experimental samples described by Ge et al.’s study was used [17]. Briefly, the samples were placed in 50 mL NaOH (0.01 M, pH 9.5) and allowed to stand at 4 °C for approximately 12 h. The residual trypsin activity of samples was assayed using BAPNA (0.6 mg/mL in 0.05 M Tris-HCl buffer, pH 8.2) as a substrate to determine their trypsin inhibitory activities. A total of 5 mL of BAPNA solution, 1 mL of sample, and 2 mL of DI water were pipetted into a test tube and pre-incubated for 10 min in a 37 °C water bath. In each of the tested tubes, 1 mL trypsin solution (0.0135 mg/mL in 0.005 M CaCl2, pH 3.0) was added and immediately incubated in a 37 °C water bath for exactly 10 min. After that, 1 mL of 30% (*v*/*v*) acetic acid was added to the tubes to stop the reaction. Additionally, 1 mL of 30% acetic acid was added to a test tube before adding trypsin as the reagent blank. The mixture was centrifuged at the speed of 4000 r/min for 10 min, and the absorbance of the supernatant at 410 nm was measured with a spectrophotometer (Shimadzu, Japan). All the measurements were in triplicate. 

### 2.5. Fluorescence Spectroscopic Analysis

The structure of DBD-treated soybean TI protein was characterized by fluorescence spectroscopy from Shanghai Unico Co., Ltd. (Shanghai, China). TI was prepared into a 1 mg·mL^−1^ solution with PBS buffer as a solvent, and the sample treated with DBD was placed in a 10 mm quartz cuvette with the light transmission on all sides, and the changes in TI intrinsic tryptophan fluorescence were measured with a fluorescence spectrophotometer at an excitation wavelength of 280 nm. Fluorescence emission spectra were monitored in the range of 290–400 nm with a slit width of 5 nm.

### 2.6. Ultraviolet (UV) Spectroscopic Analysis

The structure of the DBD-treated soybean trypsin inhibitor was characterized by UV spectroscopy from Shanghai Unico Co., Ltd. (Shanghai, China). The TI was prepared into a 1 mg·mL^−1^ solution with phosphate-buffered saline (PBS) buffer as a solvent, and the samples were placed in a 10 mm quartz cuvette for determination, and the UV scanning wavelength was 240–320 nm.

### 2.7. Circular Dichroism (CD) Spectroscopic Analysis

The secondary structure of soybean TI after DBD treatment was determined by a far-ultraviolet circular dichroism spectrometer from JASCO Co., Ltd. (Tokyo, Japan). With a TI concentration of 0.1 mg·mL^−1^, far-UV spectral scans were performed in the range of 190–260 nm using far-UV CD spectroscopy. The scanning conditions were 25 °C ± 0.5 °C, the scanning speed was 120 nm/min, the response time was 2 s, the resolution was 0.5 mm, and the sensitivity and bandwidth were set to 1 nm.

### 2.8. Caco-2 Cell Culture and Viability Assay

Caco-2 cells were cultured in Dulbecco’s modified Eagle’s medium (DMEM) replenished with 10% 35-081-CV fetal bovine serum (Corning Inc., New York, NY, USA) at 37 °C in an atmosphere of 5% CO_2_ in saturated humidity. Specific methods referred to Zhang et al.’s research [18].

In this experiment, different concentrations (0.00001–10 mg·mL^−1^) of TI and their effect on the viability of Caco-2 cells were first determined, and then the concentration with the most significant difference was selected, and PBS was used to dissolve TI at this concentration. Additionally, DBD passivation treatment was performed to explore cell viability.

### 2.9. Animal Treatment

Four-week-old male C57BL/6J mice were kept in the SPF animal room (22 ± 2 °C, 40~70% relative humidity, 12 h light/12 h dark cycles) of the Agricultural Product Quality Supervision, Inspection and Testing Center (Beijing, China) of the Ministry of Agriculture. After one-week acclimatization, they were randomly divided into four groups (5 mice per group): (1) mice received sterile water at regular intervals of 10 mL·kg^−1^ (B.W.) as a control group; (2) mice administered with 10 mg·mL^−1^ TI solution of 10 mL·kg^−1^ (B.W.) as the TI group; (3): mice received with 10 mg·mL^−1^ DBD-TI solution after DBD plasma treatment of 10 mL·kg^−1^ (B.W.) as the DBD-TI group. The experimental group was given normal food and water except for gavage. The animals were weighed every two days and changes in body weight were observed. After 20 days, all animals were fasted for 8 h and then sacrificed. The livers, kidneys, testis, and spleen were weighed.

Serums of mice were collected, and a biochemistry analyzer (Thermo Fisher Scientific Inc., Waltham, MA, USA) was used to determine the alkaline phosphatase (ALP), aspartate aminotransferase (AST), alanine aminotransferase (ALT), creatinine (Cre) and urea nitrogen (Urea). The hematoxylin and eosin (H&E) (Sigma-Aldrich Inc., Saint Louis, MO, USA) stainings of the liver, spleen, kidney, pancreas, stomach, duodenum, jejunum, and ileum were in reference to our previous method [19].

The experimental design was approved by the Animal Ethics Committee of the Inspection and Testing Center for Genetically Modified Organisms of the Ministry of Agriculture (AW42901202-4-6).

### 2.10. Effect of DBD Passivation Technology on Soymilk Quality

The untreated soymilk in 2.2 and DBD-treated soymilk in 2.3 were used as a control group and DBD group, respectively.

The PH and conductivity of the treated soymilk samples were measured by a METTLER TOLEDO pH meter (Zurich, Switzerland) and an electrical conductivity meter from Shanghai Lichen Bangxi Instrument Technology Co., Ltd. (Shanghai, China). The color of two kinds of soymilk samples was determined by a color difference meter (Shanghai Jiao-lei Automation Technology Co., Ltd. Shanghai, China) and the values of the L*a*b* color system were recorded. Determination of amino acid content in soybean milk was according to the Chinese Standard in English/GB 5009.124-2016. The soymilk sample (0.5 mg) was moved into the hydrolysis tube, then 6 M hydrochloric acid solution was slowly added, frozen for 3~5 min, sealed in a vacuum, hydrolyzed at 110 °C for 22 h in the electrothermal blast incubator, and then cooled to room temperature. After cooling, the mixture was filtered, the volume set to 50 mL, decompressed and evaporated, and a sodium citrate buffer solution was added to dissolve, oscillate and blend, filter, and determined by the amino acid analyzer (MembraPure GmbH, Bourdenheim, Germany). The viscosity of soymilk samples was measured by a rheometer (TA INSTRUMENTS, New Castle, NSW, USA) with the change of shear rate. The lamina with a diameter of 40 mm was used for the clamp, and the lamina space was 500μm.

### 2.11. Statistical Analysis

The results are presented as means ± SEMs. Significant differences between groups were calculated using Student’s *t*-test, which was considered significant at *p* < 0.05. The GraphPad Prism 8.0 package was used for graphing. 

## 3. Results and Discussion

### 3.1. Effect of DBD Plasma Treatment on TI Activity

The schematic diagram of the DBD plasma treatment system is shown in Figure 1A. As shown in Figure 1B, compared with the TI activity of unsoaked soybean flour (32.09 mg·g^−1^), after soaking treatment, the TI activity did not change, and the difference was not significant. However, TI activity decreased from 32.09 mg·g^−1^ to 24.85 mg·g^−1^ after 1 min of DBD treatment (*p* < 0.05). After treatment for 3 min, TI activity decreased to 13.25 mg·g^−1^ (*p* < 0.01). With time prolonged to 5 min, TI activity decreased to 5.21 mg·g^−1^, and the inactivation rate reached 84%. The inactivation of trypsin inhibitors in soybean can be effectively induced by DBD plasma treatment in a time-dependent manner, which was consistent with the results of previous research [16]. The scientific reason for TI activity reduction is that under the action of an electric field, active particles bombard proteins and transfer energy to the surface of protein molecules, which may promote the breaking of chemical bonds on the surface of proteins [20]. The collision between particles produces many free radicals, which can oxidize amino acids on the protein surface or introduce active sites on the protein surface, which may also damage the chemical bonds on the protein surface, leading to protein disintegration [21]. Therefore, DBD plasma can be used as an effective non-heat treatment method to passivate anti-nutritional factors in soybean.

### 3.2. Effect of DBD Plasma Treatment on the Structure of TI Protein

By studying the changes in fluorescence emission spectra of TI protein after DBD treatment, we can understand the changes in the molecular conformation of the whole TI protein. Generally, the intrinsic fluorescence spectra of proteins are mainly attributed to the fluorescence emission of aromatic amino acid residue. The peak intensity of maximal fluorescence was shown a significant decrease after DBD treatment at different times, which may be due to the quenching phenomenon in TI protein solution under DBD treatment; this resulted in a rapid decrease in fluorescence intensity (Figure 2A). The result is consistent with that reported by Ekezie et al., who found an obvious decline in the fluorescence intensity of myofibrillar proteins after non-thermal plasma treatment, alongside an apparent red shift in the maximum emission wavelength [23]. The decrease in the fluorescence intensities confirmed that plasma-inherent reactive species could modify the microenvironment of Trp residues due to oxidation [24]. The change illustrated that modification of TI conformation led to aggregation or reassociation reactions.

Additionally, the conformational changes in TI induced by DBD treatment were also deduced using UV absorption spectroscopy. As mentioned earlier, for native proteins, the UV absorption spectra are predominately determined by Trp, Try, and Phe present in the side chain groups [25]. In different microenvironments, the change in the UV absorption spectrum of chromophores is the main reason for the change in protein conformation [26]. As shown in Figure 2B, the maximum UV absorption value of TI protein increased from 0.268 to 0.745 and 0.86 after DBD treatment for 1 min and 3 min, respectively, and decreased to 0.54 after DBD treatment for 5 min. The main reason for the increased UV absorption spectrum of proteins is that the protein molecules are unfolded to expose the aromatic amino acid residues embedded in the molecules. It could be inferred that after 1–3 min of DBD treatment, the protein continued to expand, and at 5 min, the protein formed a polymer, and the previously exposed amino acid residues formed a polymer and were re-entrapped, resulting in the decrease of absorbance [27].

CD spectral analysis is a conventional method used to ascertain changes in the secondary structure of proteins [23]. Several types of research have been conducted with CD spectral analysis to study the relationship between the functions and the higher order structures of proteins since it is discovered that CD spectral reflects faithfully the secondary structures of proteins including α-helix, β-sheet, β-turn and random coils [28,29]. As represented in Figure 2C, with the increase of DBD treatment time from 1 min to 5 min, the negative peak intensity of the curve at 200 nm gradually decreased, especially at 5 min; the curve tended to level, and the results showed that the content of irregular crimp in the secondary structure of TI decreased significantly; and the corresponding content of β-helix and β-rotation angle increased (Figure 2D). Such changes might be due to the oxidation of amino acid side chains, which had been confirmed by the determination of carbonyl and sulfhydryl groups in Sharafodin et al.’s study [12]. The disulfide bonds naturally present in the protein structure have a crucial influence on the stability of the protein. Their dissociation can destabilize the original structure of the protein, thus persuading the protein to adopt a β-sheet structure more easily. Meng et al. also reported variations in secondary structures of milk proteins exposed to gamma irradiation [30]. Furthermore, Zhang et al. showed that the conformational structure of lactate dehydrogenase changed after 300 s of atmospheric pressure DBD plasma treatment, with a 12.60% decrease in α-helix content and a 10.40% increase in β-sheet content, and their explanation for this was that the active substance and UV photons changed the structure of amino acids, including disruption of peptide bonds, rearrangement, and modification of amino acids, which led to changes in secondary structure [31]. Taken together, DBD treatment destroys the secondary structure of TI in soybeans, leading to the inactivation of TI. However, there are many factors affecting the DBD plasma treatment method, such as voltage, time, oxygen concentration, etc. Due to the equipment conditions, only the time factor was investigated in this study. The influence of other factors on DBD plasma passivation of TI still needs to be further investigated.

### 3.3. Evaluation of the Safety of TI and TI Degradation Products Treated with DBD

To assess the toxicity of TI itself and the safety of degradation products of TI after DBD treatment, we performed Caco-2 cell viability assays in vitro. The result showed that when the concentration of TI was less than 0.1 mg·mL^−1^, there was no obvious toxicity to Caco-2 cells, but with the increase of the concentration of TI, the activity of Caco-2 cells decreased to 60% (*p* < 0.001) when the concentration of TI reached 1 mg·mL^−1^. At a concentration of 10 mg·mL^−1^, cell survival was only 40% (*p* < 0.001) (Figure 3A). Therefore, a concentration of 1 mg·mL^−1^ was chosen to study the cytotoxicity of DBD-treated TI. The results showed that after DBD treatment, the cytotoxicity of TI degradation products decreased and cell survival reached 75%, which was significantly different from the control group (Figure 3B). Previous research reported that 20 min high-voltage atmospheric plasma treatment of aflatoxin B1 showed a significant cytotoxicity reduction, and the loss of toxicity may be due to aflatoxin B1 C8–C9 double-bond degradation during the treatment [32]. Soybean trypsin inhibitors have been reported to be cytotoxic to cervical adenocarcinoma, HeLa, and hepatocellular carcinoma, HepG2, human cell viability [33]. In our study, TI itself reduced the viability of Caco-2 cells, whereas the DBD-treated TI degradation products were not detrimental to the cells due to the disruption of the structure of trypsin inhibitors.

### 3.4. Evaluation of the Safety of TI and TI Degradation Products Treated with DBD In Vivo

The effect of TI on the body weight of DBD-treated mice is shown in Figure 4A. After 20 days of administration of untreated TI, the final body weight of mice increased by only 2 g with a stable trend. In contrast, with DBD-treated TI, the body weight of mice started to decrease slightly, and then showed a continuous and stable growth trend, which was consistent with that of the control group. On the sixth day, the difference in body weight gain between the TI and TI-DBD groups was very obvious. The weight loss of mice in the TI group may be based on the fact that TI itself affects the digestion and absorption of proteins. By calculating the ratio of the major organs (liver, spleen, kidney, testicle) to body weight, it can be seen that the ratios of the three groups were similar; there was no significant difference between the experimental groups (*p* > 0.05) (Figure 4B). Serum levels of ALP, ALT, and AST can be used as indicators to assess the liver function in mice. Urea and CRE can be used as indicators to evaluate the renal function of mice. As seen from Figure 4C–G, no significant differences in liver and kidney function indices were observed after the ministration of TI to mice compared with the control group (*p* > 0.05).

After the administration of TI in the positive group, large areas of hepatocyte necrosis with numerous inflammatory cell infiltrates were seen locally in the tissue, as shown by the yellow arrow (Figure 4H). There was a slight abnormality in the spleen tissue and a large amount of neutrophil in the red marrow, as shown by the yellow arrow. Individual glomerular structural incompleteness and loss of intra-glomerular structure were seen in the kidney tissue, as shown by the yellow arrow; the stroma was mildly hemorrhagic and without inflammation, as shown by the red arrow. No lesions were found in the pancreatic tissue. No pathological changes were observed in the liver, spleen, and kidneys of mice with degradation products after DBD-treated TI (Figure 4H). Broilers suffered from poor growth and decreased performance due to pancreatic hypertrophy when excessive TIs were present in broiler feed [34]. An initial study reported that a short-term diet of soybean trypsin inhibitors caused pancreatic hypertrophy and hyperplasia in rats [35]. In this research, it was found that TI did not cause damage to the pancreas of CD-1 mice, but to the liver and spleen tissues. Perhaps the difference in this phenomenon is related to the time of the experiment, the dose, or the species of mice.

In the TI group, the gastric tissues of the mice showed lesions with indistinct epithelial cell structure in the mucosal layer and partial detachment, as shown by the yellow arrow; the submucosa was severely edematous, as shown by the red arrow (Figure 4I). Individual inflammatory cell infiltrates were seen in the duodenum, as shown by the yellow arrow. In the jejunum, the localized mucosal epithelial detachment was seen, as shown by the yellow arrow; the mucosal crypt was intact with localized massive inflammatory infiltration, as shown by the red arrow; the submucosa was not edematous and inflammation was visible, as shown by the black arrow. Individual inflammatory cells were seen in the ileum, as shown by the yellow arrow; some cells were necrotic, as shown by the red arrow, and some cells were necrotic, as shown by the red arrow. After administration of DBD-treated TI to mice, no pathological changes were observed in the gastric tissues except for the unclear structure and partial shedding of epithelial cells in the mucosal layer (yellow arrow), and no lesions were observed in the duodenum, jejunum, and ileum (Figure 4I).

TI could cause moderate or severe lesions in the liver, spleen, kidney, stomach, and small intestine tissues of the mice, while DBD treatment significantly alleviated the damage to the liver, kidney, and gastrointestinal tract caused by TI. In this study, we innovatively evaluated the safety of the TI itself and the degradation products of the TI after DBD passivation in vivo and in vitro.

### 3.5. Effect of DBD Passivation Technology on Soymilk Quality

The activity of TI was significantly decreased in soymilk treated with DBD for 5 min (Figure 5A), indicating that DBD plasma can be used as an effective non-heat treatment method to passivate protein anti-nutritional factors. Among the constraints generally evaluated for assessing processed food quality, pH plays a critical role since it impacts the processing specifications necessary for developing safe products and is closely monitored in almost all processing operations. Compared to the untreated soymilk, the pH of soymilk treated with DBD plasma decreased slightly without significant difference (Figure 5B). The same was true for conductivity (Figure 5C). Pankaj et al. reported no changes in the pH of white grape juice processed by plasma [36]. Elaine Porto et al. also reported the pH after plasma processing did not differ significantly compared to the non-processed coconut water (control sample) [37]. However, Almeida et al. detected a small decrease in pH values of prebiotic orange juice after plasma processing [38]. It can be speculated that the impact of plasma on the pH value of complex food matrices is often modified by several aspects such as processing time, treatment intensity, food type, buffering capacity, and physiological activities of the living tissues [39].

The effect of DBD treatment on the fluid properties of soymilk was shown in Figure 5D. The relationship between shear stress τ and shear rate γ was linearized and regression analysis performed to obtain the values of K and *n*. The consistency coefficient K and flow characteristic index *n* of soymilk treated by plasma did not change much (Appendix A), indicating that DBD treatment had little effect on the rheological characteristics of soymilk, and the original rheological properties of soymilk could be maintained. In addition, the color change of soymilk was detected, and there was no significant change in the L value, a value, and b value (Figure 5E–G). By color analysis, DBD plasma had only a slight effect on the color of soymilk, which might be due to the slight heat generated during the discharge process. Subjecting their whey protein isolate solution to atmospheric DBD plasma treatment, Segat et al. observed a significant increase in yellowness, attributing this to the limited non-enzymatic browning reaction between reducing sugars and amino acids that led to the production of early stage products of the Maillard pathway [40]. Nevertheless, in our research, plasma treatment was not able to change the color of the soymilk because the reactive substances, such as lipids, carbohydrates, and pigments, were not in sufficient quantities to trigger the Maillard reactions to a measurable extent [41]. Moreover, DBD plasma treatment could maintain the amino acid composition and content of soymilk unchanged based on inactivating the anti-nutritional factors of soymilk (Appendix A).

## 4. Conclusions

In this study, DBD plasma treatment inactivated TI and destroyed the secondary and tertiary structures of TI. The safety evaluation analysis of in vitro and in vivo experiments showed that the degradation products of TI after DBD plasma treatment were opposite to the results of TI, as shown by the reduction of the evaluation of the safety of cells, the great improvement of body weight in mice, and the alleviation of the liver, kidney and digestive tract damage caused by anti-nutritional factors. The quality analysis of soymilk showed that DBD plasma treatment was still effective in reducing the TI content in soymilk, and had no effect on the acidity, alkalinity, conductivity, color, and amino acid composition of soymilk with no statistical difference. Overall, this study presents a potential application of plasma technology in the food processing industry to reduce the anti-nutritional properties of soybeans.

## Figures and Tables

**Figure 1 foods-11-04017-f001:**
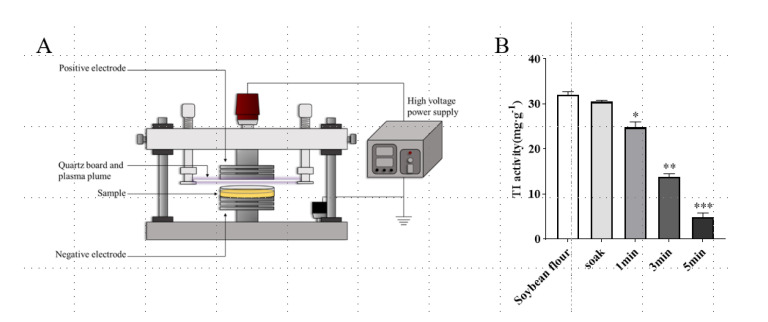
Effect of DBD plasma treatment on TI activity of soybean. (**A**) Schematic diagram of DBD plasma treatment system [22]. (**B**) Soybean flour was treated by soaking or by DBD plasma at 33.8 kV for different time intervals. Data are shown as mean ± SEM (*n* = 3). * *p* < 0.05, ** *p* < 0.01, *** *p* < 0.001 compared with the soybean flour group.

**Figure 2 foods-11-04017-f002:**
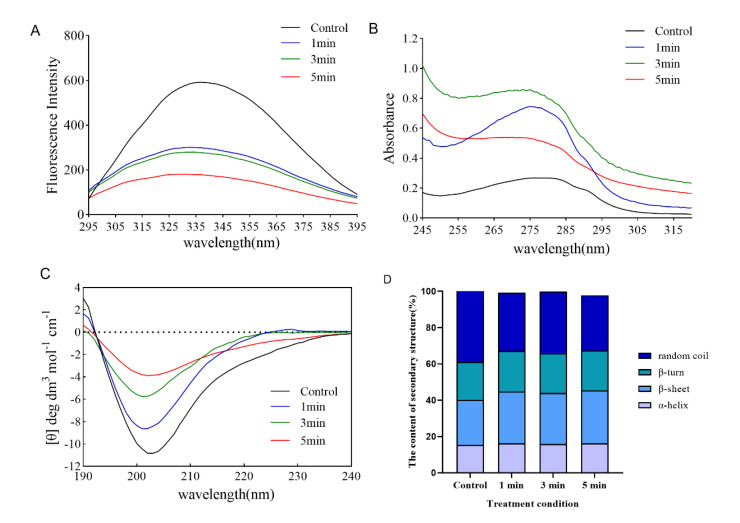
Effect of DBD plasma treatment at different times on the structure of TI protein. (**A**) Fluorescence spectroscopy, (**B**) UV spectroscopy, (**C**) CD spectroscopy, and (**D**) the content of the secondary structure are presented.

**Figure 3 foods-11-04017-f003:**
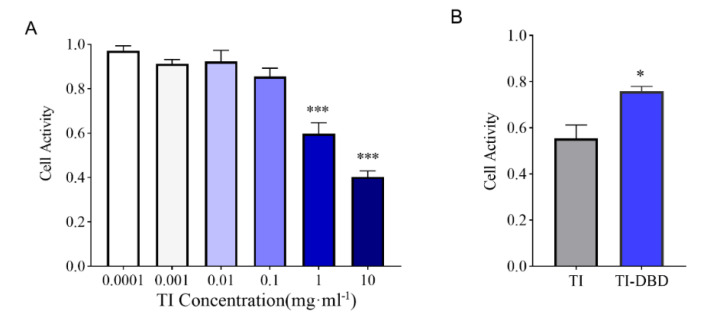
Evaluation of the safety of TI and TI degradation products treated with DBD on Caco-2 cells. (**A**) Cell viability of Caco-2 cells treated with different concentrations of TI (*n* = 3). (**B**) Cell viability of 1 mg/mL TI treatment and TI degradation products after DBD treatment (*n* = 3). Data are shown as mean ± SEM. *** *p*< 0.001 compared with the group with a TI concentration of 0.0001, * *p* < 0.05 compared with the group of TI.

**Figure 4 foods-11-04017-f004:**
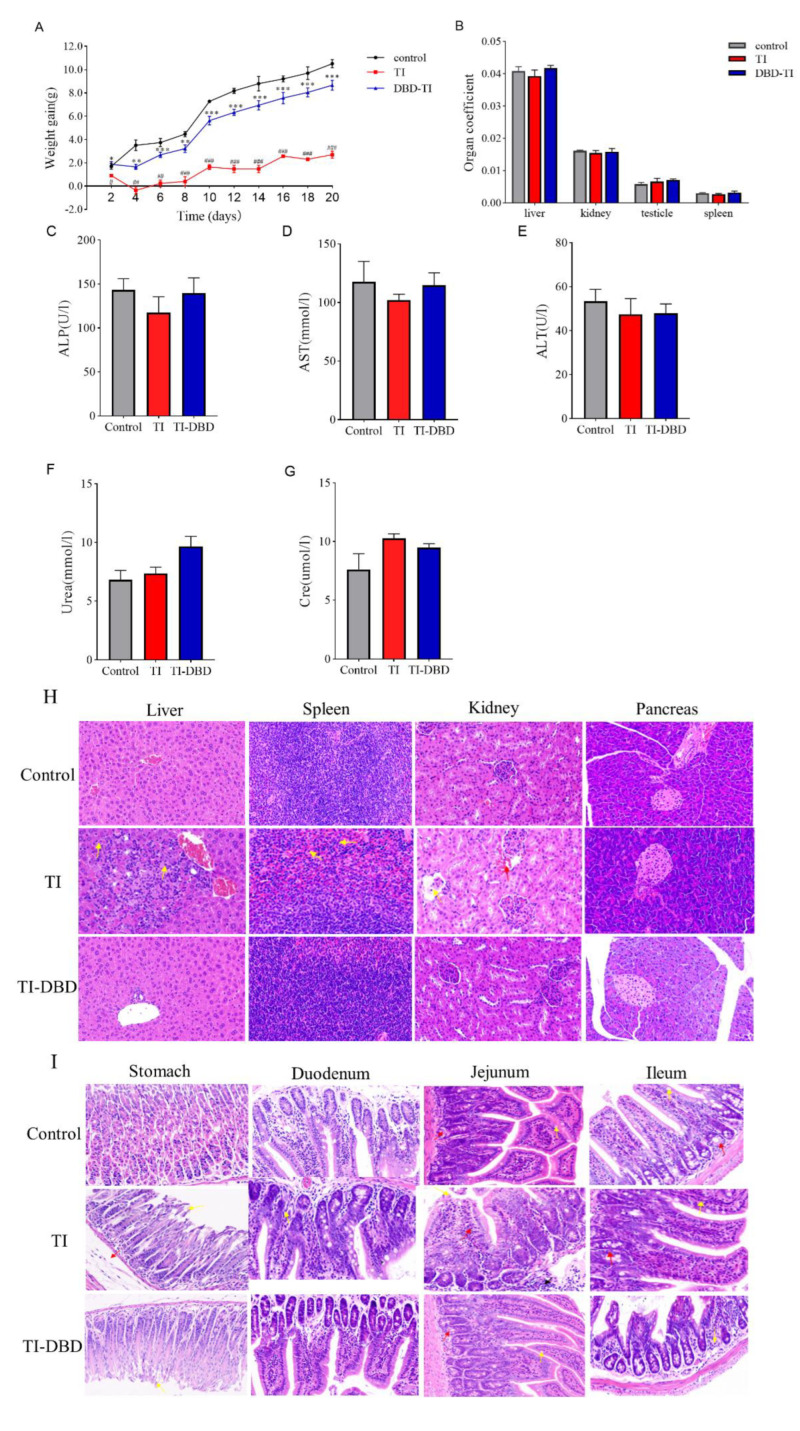
Evaluation of the safety of TI and TI degradation products treated with DBD in vivo. (**A**) Body weight gain curve (*n* = 5). (**B**) Organ coefficients of the liver, spleen, kidney, and testicle (*n* = 5). (**C**–**G**) Serum ALP, AST, ALT, Urea, and Cre levels in mice (*n* = 5). (**H**,**I**) Representative images of visceral tissues stained by H&E staining in mice (*n* = 5). The scale bar is 50 um. Data are shown as mean ± SEM. * *p* < 0.05, ** *p* < 0.01, *** *p* < 0.001 compared with the control group. # *p* < 0.05, ## *p* < 0.01, ### *p* < 0.001 compared with the TI group.

**Figure 5 foods-11-04017-f005:**
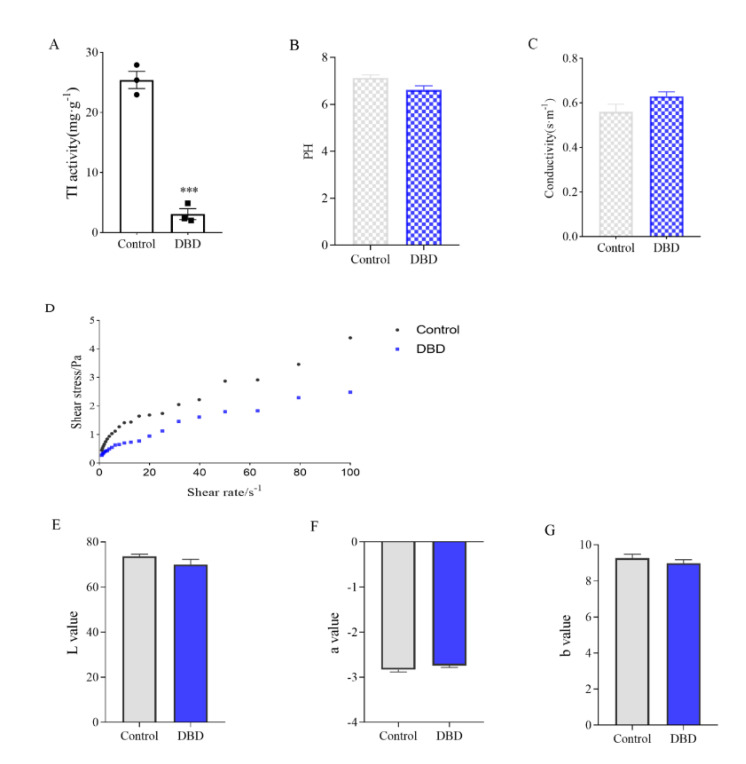
Effect of DBD passivation technology on soymilk quality. (**A**) TI activity of soymilk. (**B**) PH values. (**C**) Electrical conductivity. (**D**) Rheology curve. (**E**–**G**) Chromaticity value. Data are shown as mean ± SEM (*n* = 3). *** *p* < 0.001 versus the control group.

## Data Availability

The data are available from the corresponding author.

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
