# Peer review of "Inactivation of Soybean Trypsin Inhibitor by Dielectric-Barrier Discharge Plasma and Its Safety Evaluation and Application"

_foods, 2022, doi:10.3390/foods11244017_

Round 1

Reviewer 1 Report

Hi dear

This article "Inactivation of Soybean Trypsin Inhibitor by Dielectric-barrier Discharge Plasma and Its Safety Evaluation and Application” was revised and has a novelty after consideration of the following comments.

Title: It is perfect and comprehensive.

Abstract:

·       The type of statistical design used in this research should be mentioned.

·       Please explain your results as detail and include statistical levels for results.

·       Please include “dielectric-barrier discharge (DBD) plasma treatment” characterizations.

Keywords: Please choose keywords other than the main words of the title. In this case, other researchers can find your article by searching a wide range of words through databases. I propose another keywords as the follow:

dielectric-barrier discharge plasma; soybean trypsin inhibitor; inactivation; safety evaluation; soymilk quality, Human colorectal cancer cells (or Caco-2 cell viability), Physicochemical properties, Rheological properties

Introduction:

·       Please consider more new application cold plasma in food industry. 

·       Please note the treatments as detail in the final of introduction paragraph.

Materials:

·                 Please write materials as Company Name (City, Country), especially for chemical analysis assessment which used in the study.

Methodology:

·       Line 120-123: please include cold plasma parameters used for soymilk treatments.

·       Line 124-126: Please explain as the other methodology titles as detail protocol.

·       Line 175-189: The way of expressing the method of measuring macronutrients and other parameters has a scientific flaw. Please take help from the following article for the correct way of expressing it, so that the standard number of the working method should be clearly stated (https://doi.org/10.1590/fst.60820).

·       Line 181: Please added the other parameters of color i.e., browning index, ΔE and etc. 

 Results:

·       All Tables and Figures: The alphabetical statistical letters for the means should all be modified such that the greatest number has the letter a and as the numbers go lower, letters b, c etc.

·       Fig 2D: Please statistical comparison will be done.

·       Fig 3: please correct it as the above-mention suggestions.

·       Fig 4: The statistical comparison was not done appropriately.

Discussion:

Discussion text must grammar improve and in some cases it is very weak and maybe there is no discussion at all.

Conclusions:

Conclusion ought to be comprehensive and concise in detail, especially.

References: It is OK.

The article has many flaws in express and concept of English, it is suggested to be revised in a scientific and native way.

Reviewer 2 Report

The authors have described the dielectric-barrier plasma treatment of soybeans to inactivate trypsin inhibitors. The effects of treatment were confirmed through analyses and animal model studies. The study is well-described, and the information provided will be useful for developing advanced technologies for the processing of food materials.

Following are some of the suggestions for improving the quality of this manuscript further.

In lines 102 – 103, page 3 of 13, please specify the solution used to soak the soybeans.

Please provide the full form of ‘PBS’ in line 130.

Reviewer 3 Report

The authors investigated the effect of dielectric-barrier discharge (DBD) plasma treatment on soybean trypsin inhibitor (TI) content and its structure. In particular, it is an interesting study in that it showed that DBD plasma affected the structural transformation of TI and suggested a method to remove TI from soybean products by non-thermal treatment.

However, the authors interpreted the results as DBD plasma selectively affecting TI.

1) If so, the hypothesis that DBD plasma selectively reacts with TI molecules and the reaction mechanism should be presented. (In 3.1 section)

2) It will be easier for the reader to understand if a photograph or diagram of the sample treatment process by DBD plasma is added to Section 3.1.

3) The effect of DBD plasma treatment on beneficial nutritional components of soybean or soybean milk should be evaluated.

In particular, changes in the content of major carbohydrates, fats, vitamins, and potassium should be quantitatively analyzed (such as ICP, GC, and HPLC).

4) Some typos, such as spacing, have been found. Carefully review the entire manuscript.

Round 2

Reviewer 3 Report

The authors have well corrected the manuscript according to the comments.

The quality of the manuscript has been improved.

In the future, design experiments using high-resolution quantitative analytical equipment and draw conclusions based on accurate evidence.